# Affordable Robotic Mobile Mapping System Based on Lidar with Additional Rotating Planar Reflector

**DOI:** 10.3390/s23031551

**Published:** 2023-01-31

**Authors:** Janusz Będkowski, Michał Pełka

**Affiliations:** 1Institute of Fundamental Technological Research Polish Academy of Sciences, 02-106 Warsaw, Poland; 2Tooploox, 53-601 Warsaw, Poland

**Keywords:** automatic calibration, solid-state lidar, reshape field of view, 3D mapping, SLAM, robotic mapping

## Abstract

This paper describes an affordable robotic mobile 3D mapping system. It is built with Livox Mid–40 lidar with a conic field of view extended by a custom rotating planar reflector. This 3D sensor is compared with the more expensive Velodyne VLP 16 lidar. It is shown that the proposed sensor reaches satisfactory accuracy and range. Furthermore, it is able to preserve the metric accuracy and non–repetitive scanning pattern of the unmodified sensor. Due to preserving the non–repetitive scan pattern, our system is capable of covering the entire field of view of 38.4 × 360 degrees, which is an added value of conducted research. We show the calibration method, mechanical design, and synchronization details that are necessary to replicate our system. This work extends the applicability of solid–state lidars since the field of view can be reshaped with minimal loss of measurement properties. The solution was part of a system that was evaluated during the 3rd European Robotics Hackathon in the Zwentendorf Nuclear Power Plant. The experimental part of the paper demonstrates that our affordable robotic mobile 3D mapping system is capable of providing 3D maps of a nuclear facility that are comparable to the more expensive solution.

## 1. Introduction

Affordable 3D scanning technology is very important for autonomous mobile robots [1]. In recent years, many advantages of 3D lidar for mobile robotics, especially based on solid–state lidar, have become evident. Thus, the overall cost of such a device has been decreasing over time. An interesting research period related to the RGB–D Kinect sensor is worth mentioning [2]. From that moment on, many researchers have studied the possibility of using dense point cloud processing for many robotic applications. One dominant result is the Point Cloud Library [3]. Moreover, it is possible to build an affordable 3D lidar based on rotating 2D lidar [4]. Such approaches have been studied in mobile mapping applications [5,6,7,8]. The advantage of rotating 2D lidar is the use of industrial measurement instruments; thus, many prototypes are sufficient to cope with extreme environmental conditions. The disadvantage is related to the need to move to a relatively high mass; thus, the mechatronic design is rather expensive. Adding a mirror or prism addresses this disadvantage [9,10,11,12]. Unfortunately, an additional reflector changes the physical parameters (e.g., range) of the lidar beams.

Other types of such devices include multi–beam rotating 3D lidars and the recently announced solid–state lidars. Unfortunately, recent solid–state lidars have a limited field of view. We address this gap by introducing the results of research related to a rotating planar reflector that efficiently reshapes lidars’ field of view. Two aspects are crucial in such an approach: data synchronization and calibration. For this reason, we demonstrate our framework for solving these important aspects. The added value of our research is that we perform experiments assuming realistic conditions.

Reshaping the field of view of the depth cameras and laser scanners introduces new applications and challenges at the same time. An example of such work is presented in [13], where the authors equipped a depth camera with two angled mirrors to obtain depth perception at both the front and back of the robot. The authors proposed a calibration method based on the graph SLAM g2o framework [14]. Application information, designing, and solving graph SLAM problem are discussed thoroughly in [15,16,17,18]. An alternative solution [19] employs mirrors for adding virtual RGB–D cameras to the mapping system. The authors develop a method for virtual and real camera calibration.

Another interesting study of the design of a radially reshaped field of view is discussed in [20]. It shows great potential for new applications. The authors focused their research on the analysis of optical aspects and scan patterns, but they did not touch geometrical calibration. The authors also described reshaping the radial field of view into a new narrowed one. The problem of such calibration is not discussed [21], but we addressed it in our previous work [22], which is extended for the calibration of the 3D sensor presented in this paper. The idea of reshaping a field of view using mirrors or prisms can be implemented in several applications. Such a solution enables wider perception for unmanned aerial vehicles with a minimal increase in weight [23]. Another example is shown in [24], where such an approach is utilized in the self–driving vehicle domain.

Data calibration based on multi–view data registration is related to the SLAM [25,26] that corresponds to the so–called “chicken and egg dilemma”— the robot or other agent needs to simultaneously localize itself on the map that it is building. It is evident that by having a proper map, we can localize easier, and by having accurate localization, we can reconstruct the map. SLAM problems are divided into two groups: front–end and back–end [14]. The front–end SLAM problem is solved for a map and trajectory that is locally consistent; it consumes raw sensors data, performs feature extraction, and optimizes new poses that are added to the SLAM system. The front–end SLAM cannot adjust the trajectory and map that was obtained in the past; it only affects the most recent nodes of the trajectory. On the other hand, back–end SLAM is used to optimize the whole SLAM system. The back–end SLAM problem is formulated using the trajectory from the front–end and some other information, such as the detection of loop closure. Loop closure is detected when the robot revisits a previously mapped part of the environment.

The paper is organized as follows: Section 2 describes methodological aspects of the calibration procedure, Section 3 shows the mechanical design of the rotating planar reflector, Section 4 explains the synchronization aspects, and Section 6 presents our findings and concludes. An open–source project (example data sets and source codes) is available in [27].

The idea of reshaping the field of view with mirrors or by applying motion to the lidar sensor is common. Our contribution is an optimization method that enables one to automatically calibrate such a device, as well as the mechanical design and strategy for precise time synchronization for modern Livox lidar. Finally, we performed a comparison of the state of the art solution and validated calibrated system against the ground truth data.

## 2. Calibration

Calibration of the affordable 3D scanning system composed of a lidar and a rotating planar reflector is essential to preserve measurement accuracy compared to the original lidar. The planar reflector changes the direction of the measurement beam; therefore, it is necessary to assume the pose of its rotation axis to recalculate the measurement point (shown in Figure 1). The goal of the calibration procedure is to minimize the distance between measured points and ground truth points. Ground truth points in this paper were acquired using a Z+F IMAGER 5010 system, providing much more accurate 3D measurements than our affordable lidar. The calibration procedure assumes a modified iterative closest point approach that incorporates the intersection of a reflective plane and the beam into an optimization problem. The multi–view approach is incorporated; thus, measurements from multiple stations contribute to the calibration result.

### 2.1. Iterative Closest Point

This section describes an observation equation for building iterative closest point optimization. Equation (Equation 1) transforms point $P^{l}$($x^{l}$,$y^{l}$,$z^{l},1$), expressed in local coordinate frame, to global coordinate frame $P^{g}$($x^{g}$,$y^{g}$,$z^{g}$) by a matrix [**R**,T].
(1)$$\Psi_{\left[R,T\right]}\left(R,T,x^{l},y^{l},z^{l}\right)=P^{g}=\left[R,T\right]P^{l}$$
The iterative closest point observation equation is given in Equation (Equation 2). It is composed of residuals ${\left[\begin{array}{ccc} x^{\delta } & y^{\delta } & z^{\delta } \end{array}\right]}^{⊺}$, target values ${\left[\begin{array}{ccc} x^{gt} & y^{gt} & z^{gt} \end{array}\right]}^{⊺}$ being ground truth points, and the model function $\Psi_{\left[R,T\right]}$, incorporating state vector β into the optimization process.
(2)$$\begin{array}{c} \underset{residuals}{\underset{︸}{\left[\begin{array}{c} x^{\delta }\\ y^{\delta }\\ z^{\delta } \end{array}\right]}}=\underset{target\;values}{\underset{︸}{\left[\begin{array}{c} x^{gt}\\ y^{qt}\\ z^{qt} \end{array}\right]}}-\underset{model\;function}{\underset{︸}{\left(\begin{array}{c} \Psi_{\underset{\beta }{\underset{︸}{\left[R,T\right]}}}\left(R,T,x^{l},y^{l},z^{l}\right) \end{array}\right)}} \end{array}$$
The state vector β parameterizes a pose, in our case using the Tait–Bryan angle convention. Rotation can be parameterized in a number of ways, refs. [28,29] such as Euler angles, quaternions [30], Rodriguez [31,32,33] and Cayley formula [34], or Givens rotations [35]. Comparisons and further discussions can be found in [36,37,38,39,40]. The final optimization problem is defined in Equation (Equation 3), where there are C pairs of points (ground truth and measured) contributing to the optimization process. It is straightforward to extend this into a multi–view optimization problem.
(3)$$\min_{R,T}\sum_{i=1}^{C}{\left(\left[\begin{array}{c} x^{gt}\\ y^{gt}\\ z^{gt} \end{array}\right]-\Psi_{\left[R,T\right]}\left(R,T,x_{i}^{l},y_{i}^{l},z_{i}^{l}\right)\right)}^{2}$$

### 2.2. Intersection of a Reflective Plane and the Beam

The rotated planar reflector calibration procedure requires the extended iterative closest point observation equation. Figure 2 shows a geometrical property of the reflective plane intersection and the beam $b^{d}$. It is marked as $r^{d}$, and it relates to the measurement.

The point (x,y,z) of the intersection of lidar beam $b^{d}$ with a reflective plane $V^{pl}$ satisfies Equation (Equation 4).
(4)ax+by+cz+d=0
where
(5)$$∥\left[\begin{array}{ccc} a & b & c \end{array}\right]∥=1$$
and *d* is the distance from the origin to the rotated plane. The lidar beam has origin $b^{o}=\left(0,0,0\right)$, and the beam’s directional vector is $b^{d}=\left(b_{x}^{d},b_{y}^{d},b_{z}^{d}\right)$, $∥b^{d}∥=1$. The intersection $P^{int}$ of the beam $b^{d}$ with the reflective plane $V^{pl}$ is given with Equation (Equation 6).
(6)$$P^{int}=-\left[\begin{array}{c} b_{x}^{d}\\ b_{y}^{d}\\ b_{z}^{d} \end{array}\right]\frac{d}{V^{pl}\cdot b^{d}}$$
where (·) is the dot product. Due to the rotation of our planar reflector, the plane $V^{pl}$ is transformed using the following equation:(7)$$\left[\begin{array}{cccc} a^{r} & b^{r} & c^{r} & d^{r} \end{array}\right]=\left[\begin{array}{cccc} a & b & c & d \end{array}\right]{\left[R^{c,r},T^{c,r}\right]}^{-1}$$
where $\left[R^{c,r},T^{c,r}\right]$ transforms the planar reflector via extrinsic calibration matrix $\left[R^{c},T^{c}\right]$ and rotation via encoder angle $\left[R^{r},T^{r}\right]$, assuming the following matrix multiplication: $\left[R^{c,r},T^{c,r}\right]=\left[R^{c},T^{c}\right]\left[R^{r},T^{r}\right]$. The inverse matrix ${\left[R,T\right]}^{-1}$ can be computed with the following:(8)$${\left[R,T\right]}^{-1}=\left[\begin{array}{cc} R^{⊺} & -R^{⊺}T\\ 0 & 1 \end{array}\right]$$
Thus, the plane (planar reflector) transformation equation is formulated as follows: (9)$$\Psi_{\left[R^{c,r},T^{c,r}\right]}\left(R^{c,r},T^{c,r},a,b,c,d\right)=\left[\begin{array}{cccc} a^{r} & b^{r} & c^{r} & d^{r} \end{array}\right]=\left[\begin{array}{cccc} a & b & c & d \end{array}\right]\left[\begin{array}{cc} R^{c,r⊺} & -R^{c,r⊺}T^{c,r}\\ 0 & 1 \end{array}\right]$$
The directional vector $r^{d}$ of the reflected beam $b^{d}$ is given by Equation (Equation 10). $V^{pl}$ is the normal vector of the rotated reflecting plane.
(10)$$r^{d}=2\left(b^{d}\cdot V^{pl}\right)V^{pl}-b^{d}$$
Point $P^{l}$($x^{l}$,$y^{l}$,$z^{l}$) is given in the local coordinates of the lidar. $l^{p}=∥P^{l}∥$ is a measured distance in the local coordinates of the lidar. The distance from origin to intersection is $l^{int}=∥P^{int}∥$. Finally, measurement point $P^{r}$, after applying reflections, is given with Equation (Equation 11).
(11)$$P^{r}=-\left(P^{int}+r^{d}\left(l^{p}-l^{int}\right)\right)$$
The expected measurement point in the global reference frame is given as Equation (Equation 12).
(12)$$P^{g}=\left[R,T\right]P^{r}=\Psi^{r}\left(R,T,a^{r},b^{r},c^{r},d^{r},P^{l}\right)$$
This is used to form an observation equation assuming the reference ground truth point. Finally, the calibration optimization problem is defined as Equation (Equation 13).
(13)$$\min_{R,T}\sum_{i=1}^{C}{\left(\left[\begin{array}{c} x^{gt}\\ y^{gt}\\ z^{gt} \end{array}\right]-\Psi^{r}\left(R,T,a^{r},b^{r},c^{r},d^{r},P^{l}\right)\right)}^{2}$$

For the quantitative evaluation of the 3D lidar’s accuracy, we used ground truth data obtained by a terrestrial laser scanning Z + F IMAGER 5010 system. The reference point cloud was obtained with high precision. The tool allows one to obtain range measurement with a constant uncertainty of 1 mm with a variable of 10 parts per million. For a maximum range of 178 m, the maximum range error declared by the manufacturer is no more than 3.8 mm. Expected precision degrades with lower reflectivity. The device measures two angles of the ranging beam, both with a variance of 0.007 degrees. Underground parking is considered as the testing scenario, where five stations were used for static data acquisition (Figure 3).

Two of them were used for the calibration procedure using observation Equation (Equation 13); thus, we obtained the following parameters: $R^{2},T^{2},R^{3},T^{3},R^{c},T^{c},a^{r},b^{r},c^{r},d^{r}$. $R^{2},T^{2}$ is the pose of station 2 and $R^{3},T^{3}$ is the pose of station 3. All data from both stations contribute to $R^{c},T^{c},a^{r},b^{r},c^{r},d^{r}$. To confirm the accuracy, we performed data registration of another three stations to ground truth using an optimization problem defined with Equation (Equation 3) and fixed calibration parameters from previous calculations. Thus, for each station, the 3D lidar points $x_{i}^{l},y_{i}^{l},z_{i}^{l}$ are transformed to the ith local coordinate system $R^{i},T^{i},\;i\in 1,4,5$. In this procedure, we obtain three poses as $R^{1},T^{1},R^{4},T^{4},R^{5},T^{5}$. Next, the number of planar features was manually extracted from both the point cloud obtained from the terrestrial laser scanning and the aligned point cloud obtained by the presented system. The TLS (terrestrial laser scanning) data was used to fit a plane. The plane was fitted by performing principal component analysis on points that belong to the planar feature. The basis vector that corresponds to the smallest eigenvalue was taken as a normal vector. Next, the identified plane was used to assess the accuracy and precision of the tested system, as summarized in Table 1. One of the planes is shown in Figure 4.

Columns $\mu_{tls}$ and $\sigma_{tls}$ in Table 1 consist of parameters of normal distribution of distances to the plane. Note that the mean of the distribution $\mu_{tls}$ for TLS data is zero, which is expected since the plane was fitted to that set. The standard deviation observed is larger than the error expected from the range measurement of the TLS. That is caused by a number of factors: the planar feature might have low reactivity, or it might not be ideally planar. The map from TLS consists of multiple scans that were aligned manually using specialized software, which can further contribute to the observed value of $\sigma_{tls}$. Assuming that TLS provides accurate and precise data, detected planar features can be used to assess both the accuracy and precision of the investigated solutions. For every identified feature, the normal distribution of distances to the plane of observations is described with standard deviation $\sigma_{\mathrm{livox}}$ and mean $\mu_{\mathrm{livox}}$. Observed mean $\mu_{\mathrm{livox}}$ is significantly larger than $\mu_{tls}$ for every plane. The same is true for $\sigma_{\mathrm{livox}}$, which is significantly larger than $\sigma_{tls}$. The different values of $\mu_{\mathrm{livox}}$ for different planes show that the data provided by the investigated system are expected to include some inaccuracy. That observed inaccuracy can be caused by distortion caused by imperfect calibration or the data registration. The standard deviation $\sigma_{\mathrm{livox}}$ also varies with different features: it is significantly smaller for features that are observed closer. The cumulative metric was computed by merging all measurements and down–sampling those to obtain a consistent sampling rate. This was required since a feature that is far away has a significantly lower density of points that hit that feature.

## 3. Mechanical Design

The system is mounted on the facing side of the lidar. The lidar is assembled into a faceplate with multiple M3 screws. A BLDC motor, encoder, and axis of rotation lay on an optical axis of the lidar. The circular mirror is tilted by 52.5 degrees to the side. The mirror reflects the rays emitted by the lidar, as shown in Figure 5.

The current rotation angle is measured by an incremental, contactless encoder that consists of a stationary head and a spinning ring. The mirror, its support, and the encoder’s ring are mounted on top of the BLDC motor. The BLDC motor is a low-RPM, high-torque disc motor, without Hall’s effect sensor. The motor is assembled in the housing, which precisely orients the motor with the spinning ring against the encoder’s head. That housing is mounted to the top plate. This plate is connected to the faceplate with three pillars. A prototype is shown in Figure 6.

Currently, the system is capable of the following:Rotation of the mirror up to 200 RPM;Resolution of the encoder is 15,200, with a zero pulse;Unutilized advanced controller, which allows one to precisely control the rotation speed from 1 RPM to 200 RPM, and also enables position control.

Due to the lack of Hall’s sensor and usage of an incremental encoder with zero pulse, the device needs to perform an automatic startup procedure. First of all, the motor controller identifies the direction of the magnetic field of the motor’s rotor by applying an alternating current to its phases and observing a change in position measured by the incremental encoder. The next system performs so called homing. The mirror is spun until the encoder’s zero pulse is found. With those two steps, the system is ready to be used.

## 4. Electronic Design

The electronic design of the system consists of a microcontroller with multiple extra components utilized on a PCB (printed circuit board):Advanced motor controller integrated with on board CAN Open network;Ethernet switch;RS485 conntroller (for Livox Mid–40 sync port)

The microcontroller firmware is a program that communicates with the host computer via robust, stateless UDP (user datagram protocol). It reports, with a frequency of 1 kilohertz, a current rotation angle with the timestamp. The firmware generates a physical, electric PPS (pulse per second) signal that is used to synchronize external devices, such as the Livox Mid–40. When the internal timestamp generator changes to the next second, a short pulse (50 milliseconds) is generated on PPS output and sent to the Livox Mid–40. According to the Livox Mid–40 documentation, the device on the rising edge of the PPS signal zeroes its internal timer. This timer’s count is attached to every packet sent by the Livox Mid–40. The Livox Mid–40 has a simplified synchronization mechanism that only consumes PPS signal. More advanced lidar solutions (such as Velodyne VLP 16 or multi–lidar system with Livox Hub) also consume so–called NMEA (National Marine Electronics Association) stream. This one–way serial communication from a GNSS receiver (or a synchronization device) sends a UTC (Coordinated Universal Time) timestamp in its stream. In the Livox Mid–40, this is not available, so a solution for a robust synchronization strategy is proposed. The Livox Mid–40 and the system’s microcontroller send two asynchronous UDP streams with high frequency. A data acquisition application buffers those two streams into the following:A timely sorted set for encoder (an angle with a timestamp), fixed size, around 0.5 s of a data stream;An ordinary queue for Livox Mid–40 packets (3D points with a timestamp), with a minimum size of 10 packets, and maximum of 100 packets.

The timestamp that is reported by the system’s microcontroller is in a range from 0 to, effectively, infinity (the top plot in Figure 7).

The timestamp that is reported by the Livox Mid–40 lidar is in a range from 0 to 1 s (third from the top plot in Figure 7). The Livox Mid–40 lidar detects the rising edge of the PPS pulse (second from the top plot in Figure 7) and zeroes its timer. Effectively, the Livox Mid–40 lidar reports a fractional part of the timestamp. The major, whole part of a timestamp is reconstructed in the host’s software. An extra helper register is used. This register is loaded with the whole part of the timestamp reported by the system’s microcontroller, exactly between two PPS pulses (fractional part of timestamp equal to 0.5 s). It can be seen as an initial jump, around sample 50, in the bottom plot in Figure 7. Next, when a falling edge in the timestamp reported in the packets arriving from the Livox Mid–40 lidar is detected, the helper register is incremented. The whole process can be supervised by comparing the current whole second reported by the system’s microcontroller and the helper register value exactly between two PPS pulses (fractional part of timestamp equal to 0.5 s). The arriving packet’s full timestamp is the sum of the current value of the helper register and the fractional timestamp originally reported by the Livox Mid–40 lidar.

Synchronization software uses those timestamps to attach the Livox Mid–40 lidar’s packets to correct the value of the angle read by the encoder. This is done in a separate thread. This thread pops from the queue of the oldest packet and tries to find the best fit in the ordered set of angle–timestamp pairs.

## 5. Mapping

The mapping was done by a robot equipped with multiple mapping systems: Livox Mid–40 with rotating mirror, VLP 16, and rotated Velodyne VLP 16, as shown in Figure 8.

This robot was used for the evaluation of the proposed contribution. The robot provided two 3D data streams for further comparison. The mapping exercise was done in a mobile mapping fashion with the wheeled platform Clearpath Husky equipped with an AHRS (attitude and heading reference system) using IMU (inertial measurement unit). The wheel odometry and IMU measurements are combined using an EKF filter. Filtered odometry is used to remove the distortion caused by the movement of the robot.

The mapping algorithm aggregates the local, undistorted point cloud for a given distance and arranges its SE(3) (special euclidean group) poses as a variable node in the factor graph. Such individual point clouds are shown in Figure 9, Figure 10 and Figure 11. It can be seen that the proposed solution provides a larger range of, and similar, metric measurements compared with Velodyne VLP 16 (see Figure 12). Factor graphs are a technique for modeling high dimension non linear optimization problems [41]. They are used in modern localization, filtering, and smoothing robotic applications [42,43]. For modeling and solving our SLAM problem, a state of the art library called GTSAM was used [44]. The filtered odometry at the moment of the beginning of the aggregation is the initial guess for this node pose. The EKF solution is used to undistort the point cloud that was produced during movement. The EKF solution is obtained from the velocity measured by wheel odometry and the attitude provided by AHRS IMU. This is a simple and robust approach that can be used for slow moving platforms in an urban area. This approach is not suitable for a platform where odometry is not reliable or not available, such as surface vehicles or ground vehicles in challenging terrain. In those conditions, lidar inertial odometry should be considered [45].

Next, the factor graph is populated with factor nodes. The first type of factor is introduced by odometry; it is the relative pose reported by the robotic platform. The second consists of AHRS (attitude and heading reference system) measurements, which introduce the measurement of tilt and pitch of the robotic platform. Finally, there are a number of candidate factors that are observed. Such candidates are consecutive lidar measurements and, in particular, lidar measurements that were taken at close distances. This candidate factor includes two aggregated lidar measurements that are matched with a normal distributions transform. The algorithm allows one to perform scan registration utilizing the representation of the point cloud as a grid of Gaussian distributions. When NDT (normal distributions transform) converges, the factor candidate is added to the graph. Such NDT factors create so called laser odometry and loop closure. A diagram showing a simplified version of such a factor graph is shown in Figure 13.

The mapping algorithm utilizes data from two types of systems, as shown in Figure 14 and Figure 15. The point cloud using Livox Mid–40 with a rotated reflector utilizes a non repetitive scan pattern, which is visible as curved lines in Figure 9. On the other hand, the Velodyne VLP 16 in this configuration utilizes a repetitive pattern, which is visible as a group of points visible in horizontal bands in Figure 11. Maps built with both system looks similar, but due to their broader field of view, maps built with Livox Mid–40 with a rotated reflector contain features of the environment at a higher level of detail. The non repetitive pattern allowed us to better model the planar features that are visible in Figure 14. Figure 16 demonstrates the undistorted, aggregated point cloud built using data from the Livox Mid–40 with a rotated reflector at Zwentendorf NPP and the distance to the reference point cloud from the Velodyne VLP 16. Both point clouds were compared in the intersection shown in Figure 17. This observation is also supported by quantitative measurements, as shown in Figure 18. It can be seen that the majority of distances are below 10 cm.

The obtained calibration and accuracy were compared against the DEM (digital elevation model) shown in Figure 19. The DEM was obtained from ALS (airborne laser scanning), available publicly in the Head Office of Geodesy and Cartography [46]. In the cross section shown in Figure 20 and Figure 21, the data obtained from the designed system overlap well with the DEM; thus, we consider accuracy to be satisfactory, and the effective range is more than 100 m. This experiment shows that the major discrepancies are around 30 cm.

## 6. Conclusions

This paper describes an affordable mobile mapping system based on lidar with an additional rotating planar reflector. The results show that the proposed mobile mapping system can reach comparable accuracy to the state of the art Velodyne VLP 16 sensor and is much less expensive at the same time (the hardware and manufacturing cost of our setup, excluding labor, was approximately 50% that of the Velodyne VLP 16). The solution was evaluated during the 3rd European Robotics Hackathon at Zwentendorf Nuclear Power Plant. It is shown that the Livox Mid–40 lidar with synchronized rotated planar reflector can reach a field of view of 38.4 × 360 degrees without loss of the beam pattern. It is worth mentioning that the rotated planar reflector preserves the beneficial features of the Livox Mid–40 lidar: its large range and non repetitive scanning pattern. The active method of measurement is independent of lighting conditions in comparison to other passive methods such as stereo vision. Further development of the prototype will lead to an investigation of the device in harsh conditions, such as cold or humid weather, precipitation, fog, or smoke. These will be possible when other engineering challenges, such as such proper internal protection, can be ensured. The multi view data registration scheme is proposed, and it can be adapted for other lidar systems’ calibration. This work extends the applicability of solid state lidar since the field of view can be reshaped with minimal loss of measurement properties. Thus, it is shown that the proposed method can preserve the metric accuracy. We validated the 6 cm accuracy of the lidar and the range exceeding 150 m. We introduced an affordable mobile mapping system that can compete with state of the art sensors, such as the Velodyne VLP 16. The presented setup enables a few interesting research directions, such as benchmarking approaches in other environments (e.g., underground mining, natural environments) or other domains (e.g., building high definition maps, intralogistics, or agriculture). Our solution provides 3D data with new unique characteristics: omnidirectionality and non repetitive pattern. Those two characteristics can be beneficial in mapping alogorithms without odometry, such as lidar inertial odometry [45], lidar inertial mapping and smoothing [42], or modern approaches to ego motion estimation [47].

Future work will be related to an extension of the evaluation that was carried out in large scale open air nuclear power plant scenarios. One interesting research direction is filling the gaps in DEM with data coming from TLS. Preliminary integration of DEM with data produced by our system is shown in Figure 22. Thus, a variety of different scenarios (such as different climatic conditions, different lighting conditions) will be the core of further research.

## Figures and Tables

**Figure 1 sensors-23-01551-f001:**
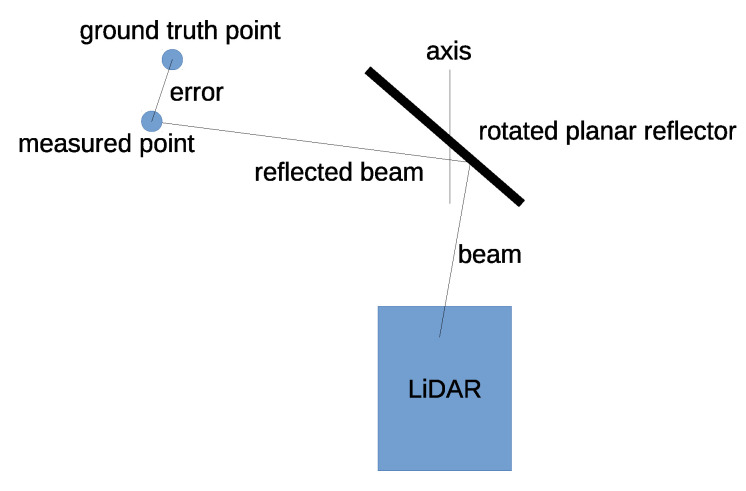
Scheme of the affordable 3D scanning system composed of lidar and rotating planar reflector. The goal of the calibration procedure is to minimize the sum of errors squared between measured and ground truth points obtained woth TLS.

**Figure 2 sensors-23-01551-f002:**
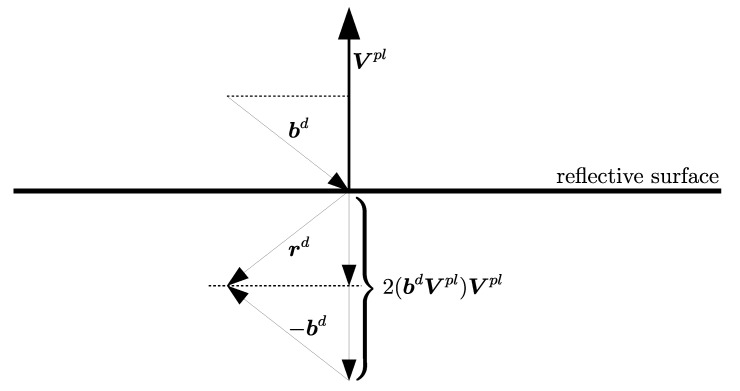
Intersection of a reflective plane and the beam $b^{d}$.

**Figure 3 sensors-23-01551-f003:**
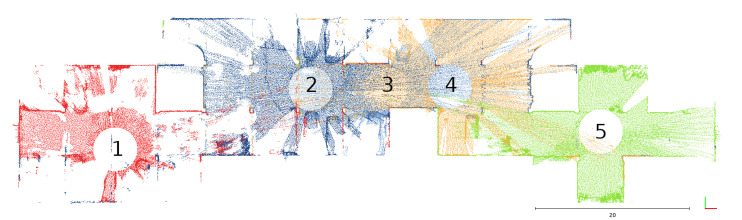
Measurement stations. Stations 2 and 3 were used in calibration. Stations 1, 4, and 5 were used in validation.

**Figure 4 sensors-23-01551-f004:**
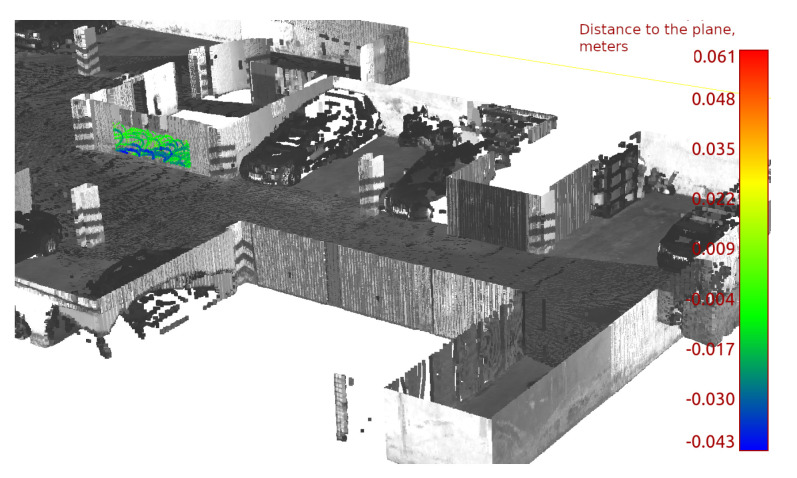
Example of planar feature. Gray scale: data from TLS. The color point cloud is the observation of the feature from the presented system. The color map shows the distance to the planar feature.

**Figure 5 sensors-23-01551-f005:**
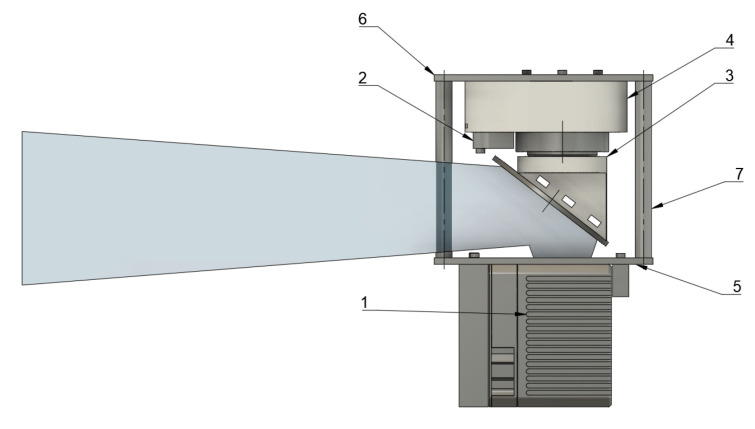
Mechanical design of mirror drive: (1) Livox Mid–40 lidar, (2) contactless encoder, (3) mirror support, (4) motor housing, (6) top plate, (5) bottom plate, (7) pillars. The resulting field of view is shown in blue. The system can be mounted in any position.

**Figure 6 sensors-23-01551-f006:**
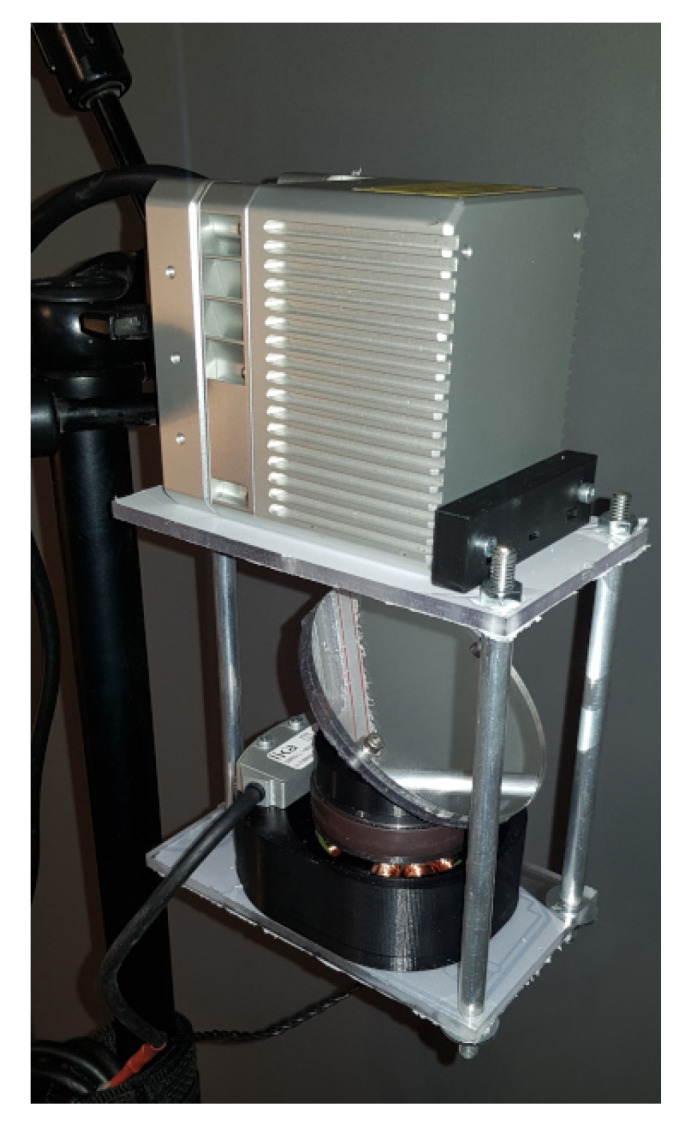
Photo of the mechanical design of the mirror drive.

**Figure 7 sensors-23-01551-f007:**
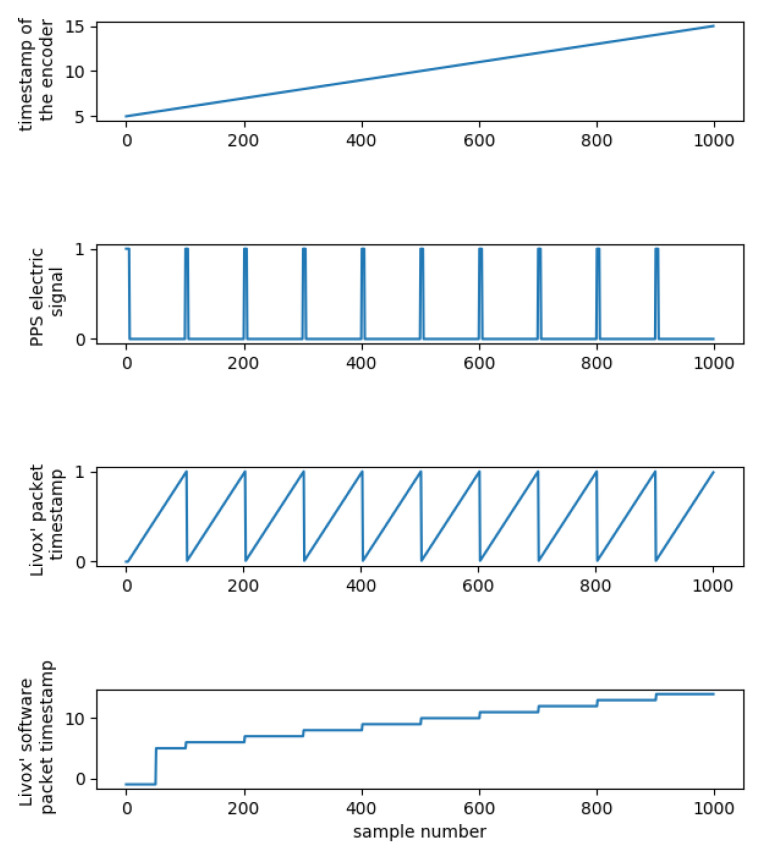
Time diagram of synchronization signals, register, and timers.

**Figure 8 sensors-23-01551-f008:**
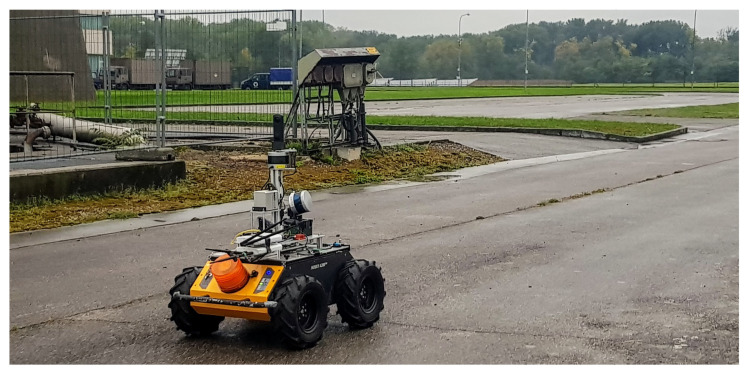
Robotic platform equipped with multiple mapping systems. This robot was used for the evaluation of the proposed contribution. The robot provided two 3D data streams for further comparison.

**Figure 9 sensors-23-01551-f009:**
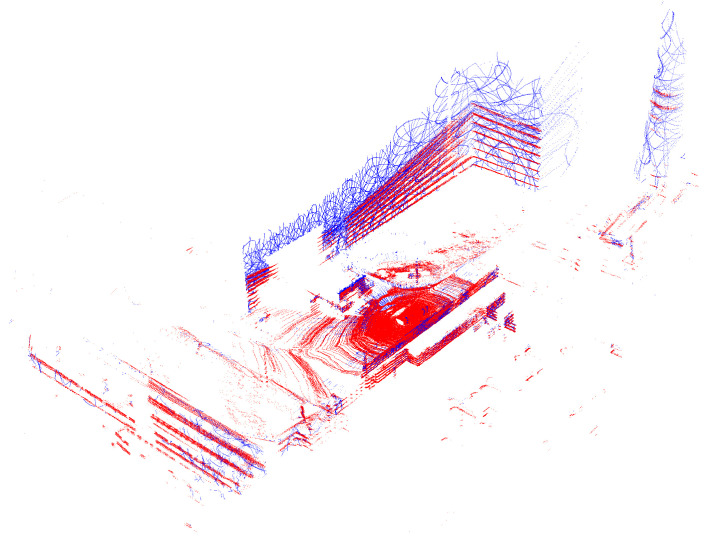
Undistorted, aggregated point clouds were built using data from the Livox Mid–40 with the rotated reflector (blue) and Velodyne VLP 16 (red) at Zwentendorf NPP. It is important to notice that our system has a larger field of view. Note the non repetitive scanning pattern produced by the rotating reflector.

**Figure 10 sensors-23-01551-f010:**
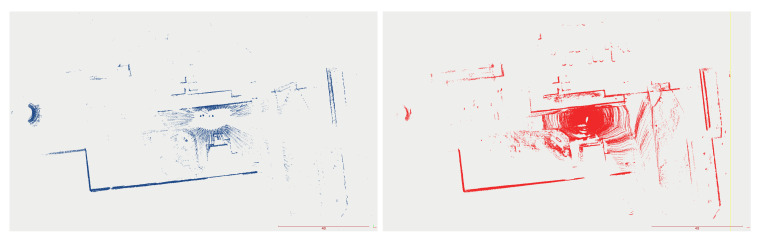
Undistorted, aggregated point clouds were built using data from the Livox Mid–40 with the rotated reflector (blue) and Velodyne VLP 16 (red) at Zwentendorf NPP. Top down view of Figure 9.

**Figure 11 sensors-23-01551-f011:**
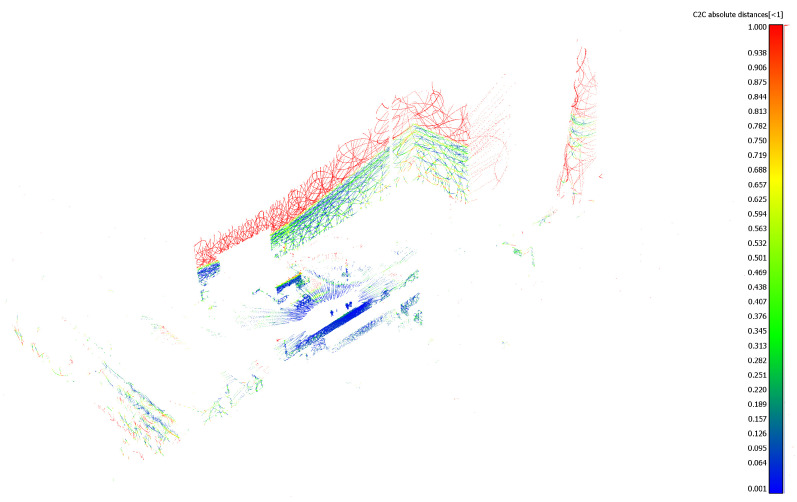
Undistorted, aggregated point cloud built using data from the Livox Mid–40 with the rotated reflector at Zwentendorf NPP. The distance to the reference point cloud from the Velodyne VLP 16 is marked by colors.

**Figure 12 sensors-23-01551-f012:**
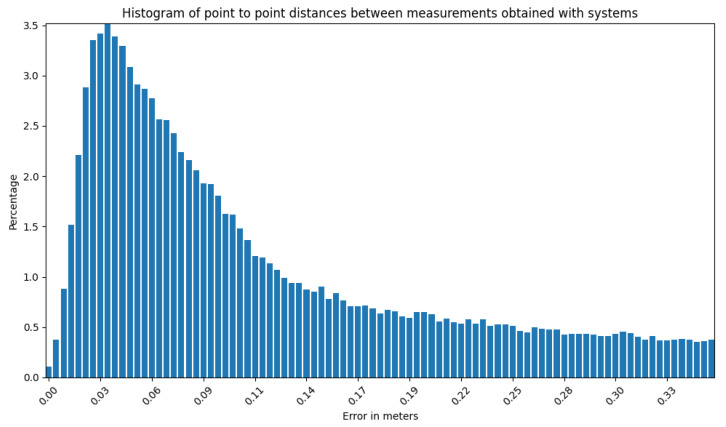
Histogram showing the distribution of distances from our system to reference Velodyne VLP 16. It can be seen the majority are below 10 cm.

**Figure 13 sensors-23-01551-f013:**
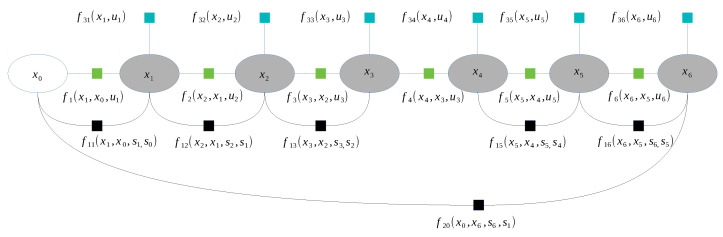
Used factor graph. Factors from $f_{1}$ to $f_{6}$ are odometry factors. Factors from $f_{11}$ to $f_{16}$ are observation factors. Factors from $f_{31}$ to $f_{36}$ are IMU prior factors. Factor $f_{20}$ is a loop closure. Note that the observation factor that connects pose $x_{3}$ and $x_{4}$ does not exist, due to failed NDT matching. The variables $u_{1}$ to $u_{6}$ are robot odometry readings. The variables $s_{1}$ to $s_{6}$ are scans taken near the corresponding poses.

**Figure 14 sensors-23-01551-f014:**
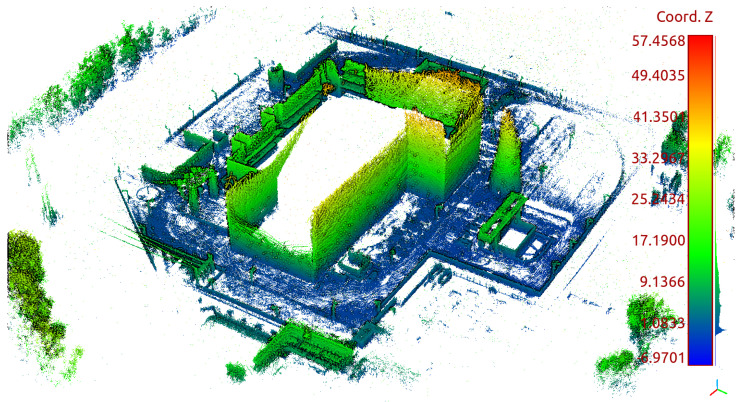
Side view of map of the Zwentendorf NPP obtained with Livox Mid–40 with rotated mirror.

**Figure 15 sensors-23-01551-f015:**
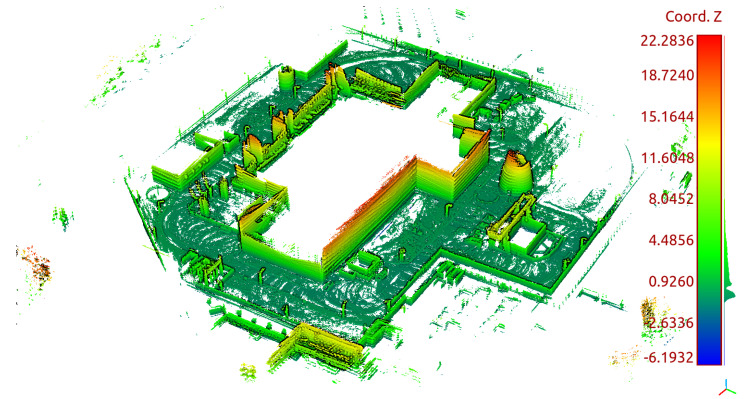
Side view of map of the Zwentendorf NPP obtained with VLP 16.

**Figure 16 sensors-23-01551-f016:**
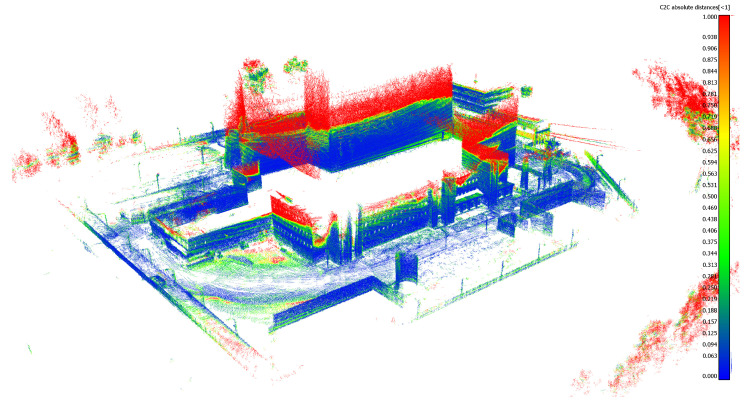
Undistorted, aggregated point cloud built using data from Livox Mid–40 with a rotated reflector at Zwentendorf NPP. Distance to reference point cloud from Velodyne VLP 16. The larger range of our solution and similar metric measurements compared with Velodyne VLP 16 can be seen.

**Figure 17 sensors-23-01551-f017:**
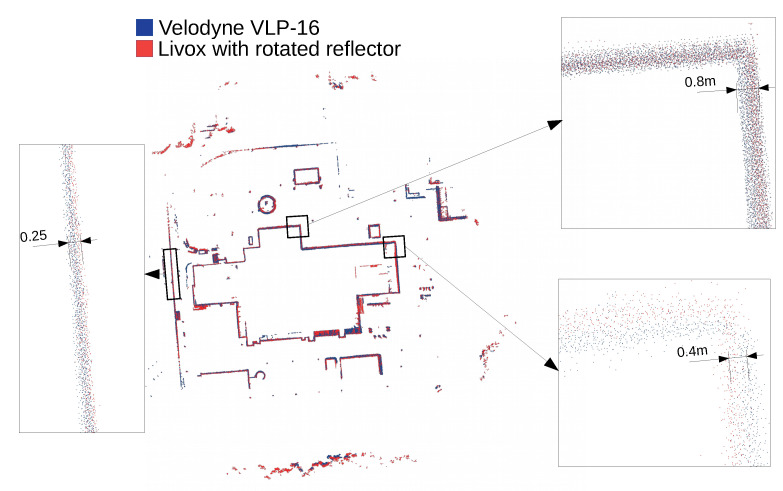
Comparison of the intersection of VLP 16 and Livox Mid–40 with rotated mirror. There is visibly no discrepancy between blue (Velodyne VLP 16) and red (Livox with rotated reflector).

**Figure 18 sensors-23-01551-f018:**
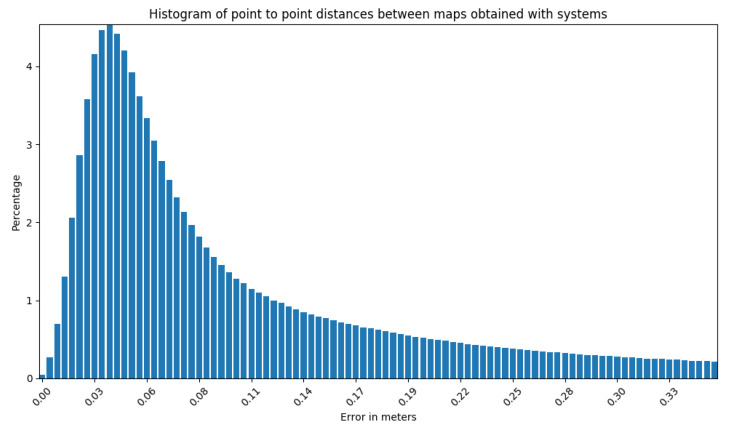
Histogram for the entire experiment showing the distribution of distances from our system to reference Velodyne VLP 16. It can be seen that the majority are below 10 cm.

**Figure 19 sensors-23-01551-f019:**
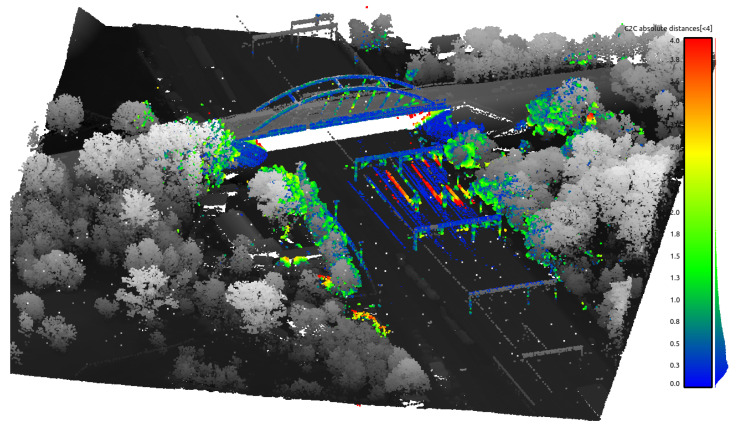
Comparison of scan obtained with designed system and DEM. DEM is given with gray scale; the obtained data from the system is given with color map. Color map encodes distance between measurement from our system and DEM.

**Figure 20 sensors-23-01551-f020:**
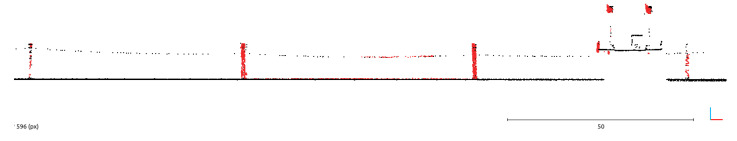
Cross section of scan obtained with designed system (red color) and DEM (black color).

**Figure 21 sensors-23-01551-f021:**
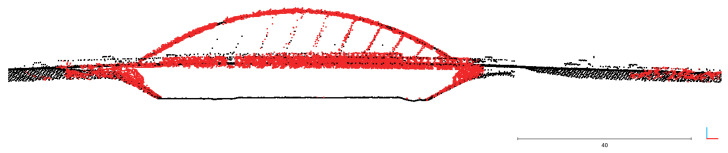
Cross section of scan obtained with designed system (red color) and DEM (black color).

**Figure 22 sensors-23-01551-f022:**
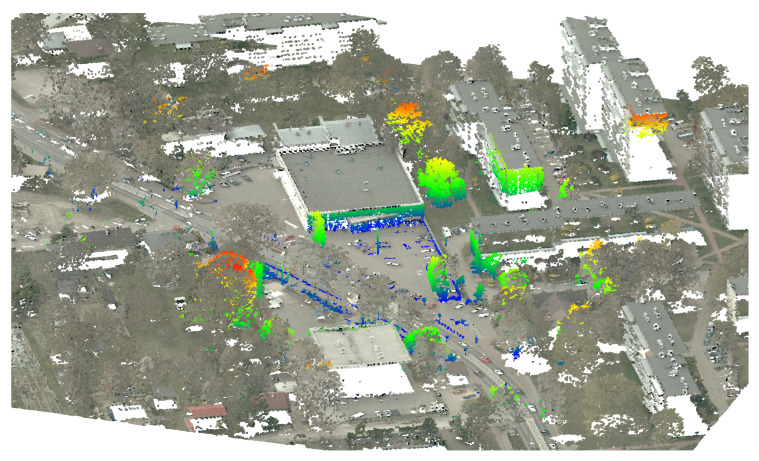
The possible use case for the proposed solution is fulfilling gaps in DEM due to ALS’ limited field of view. Points with RGB color belongs to DEM, the data with the colored height is coming from the proposed system.

**Table 1 sensors-23-01551-t001:** The normal distribution parameter of distances to fitted planar features. Each row is one feature, and each column one parameter. $N(\mu_{tls},\sigma_{tls})$ is the normal distribution approximation of distances’ distribution of points to planar features for TLS. $N(\mu_{livox},\sigma_{livox})$ is the normal distribution approximation of distances’ distribution of points to planar features for the system with rotated reflector. The last row consists of distributions’ parameters for all points and all features.

Planar Feature	$\mu_{\mathrm{tls}}$	$\sigma_{\mathrm{tls}}$	$\mu_{\mathrm{livox}}$	$\sigma_{\mathrm{livox}}$	Distance to Observation
A (wall, far)	0 mm	2.8 mm	2.3 cm	2.8 cm	55 m
B (wall, close)	0 mm	1.9 mm	0.7 cm	2.1 cm	6 m
C (floor, close)	0 mm	3.0 mm	2.6 cm	0.9 cm	4 m
D (wall, close)	0 mm	5.2 mm	2.6 cm	0.9 cm	4 m
E (wall, middle)	0 mm	1.5 mm	0.9 cm	2.4 cm	16 m
F (wall, middle)	0 mm	3.0 mm	0.4 cm	2.7 cm	35 m
Cumulative	0 mm	3.4 mm	0.9 cm	1.6 cm	−

## Data Availability

Data is available at [27].

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
