# Peer review of "Affordable Robotic Mobile Mapping System Based on Lidar with Additional Rotating Planar Reflector"

_sensors, 2023, doi:10.3390/s23031551_

Round 1

Reviewer 1 Report

1. This paper presents a 3D scanning system, which is composed of a low-cost Livox Mid-40 laser radar and a rotating planar reflector, and is mainly used for 3D mapping of robots. In addition, a multi-view data registration scheme is proposed, which is also suitable for the calibration of other lidar systems. The topic selection of the article has a strong practical significance. The article structure is clear, the innovation point is clear, to the proposed system is more detailed and complete. 2. In the process of experimental evaluation, the evaluation was only carried out in the large-scale open-air nuclear power plant scenarios, not in a variety of different scenarios (such as different climatic conditions, different lighting conditions) , it is suggested to add relevant analysis. 3. Only the accuracy of the mapping and the scanning range were evaluated, but not multiple indexes. Such as: system stability and real-time, and so on, it is suggested to add correlation analysis.

Author Response

Respond for review:

1. This paper presents a 3D scanning system, which is composed of a low-cost Livox Mid-40 laser radar and a rotating planar reflector, and is mainly used for 3D mapping of robots. In addition, a multi-view data registration scheme is proposed, which is also suitable for the calibration of other lidar systems. The topic selection of the article has a strong practical significance. The article structure is clear, the innovation point is clear, to the proposed system is more detailed and complete. 2. In the process of experimental evaluation, the evaluation was only carried out in the large-scale open-air nuclear power plant scenarios, not in a variety of different scenarios (such as different climatic conditions, different lighting conditions) , it is suggested to add relevant analysis. 3. Only the accuracy of the mapping and the scanning range were evaluated, but not multiple indexes. Such as: system stability and real-time, and so on, it is suggested to add correlation analysis.

Respond: Due to 10 short time for revision we decided to add such information in future work.

Reviewer 2 Report

A very interesting cost-effective solution, with good result quality. Could it be applicable to explore underground structures and spaces? Is the odometry enough for movement distortion filtering - as probably this solution is more sensitive to vibrations etc than devices with more static design?

Author Response

Respond for review:

A very interesting cost-effective solution, with good result quality. Could it be applicable to explore underground structures and spaces? Is the odometry enough for movement distortion filtering - as probably this solution is more sensitive to vibrations etc than devices with more static design?

Respond: We add comments in manuscript.

Reviewer 3 Report

Title: Affordable robotic mobile mapping system based on LiDAR with additional rotating planar reflector

Turnitin Result: 8%. It referred several times to "bip.put.poznan.pl". Please explain.

Result (Rating out of 60 (60=best)): 41 (68%)

(1-10(best))
Novelty: 6
Writing&Language: 2
Scientific Rigor: 8
Introduction&Literature: 9
Method&Results: 8
Discussion&Conclusion: 8

In short: The authors present a self-build lidar system for SLAM.

Overall:
In total the authors show in depth knowledge of ongoing research and development and propose a valuable idea for a low cost lidar system for SLAM. I am missing a cost evaluation though which should be provided to emphasise how cheap the system really is. Furthermore, the language needs to be improved significantly to allow easy readability. Please ask a native speaker to edit your manuscript or use a professional English editing service.

Please use "lidar" not "LiDAR" or anything else as "lidar" is a word in its own right based on the Oxford English dictionary and Merriam Webster Dictionary. (compare: radar)

The order of the introduction is a bit confused. You should state what current research is out in the area, explain relevant concepts that are very important for your paper or unknown to the target audience of the journal and then come to the problem that you want to solve and why it needs solving. You however have the aim somewhere in paragraph 2. Please rethink the order of paragraphs and also link them or easier transition for the reader between those.

Detail Comments:
page 5 line 108: You mention plane fitting without mentioning which algorithm you use. Please specify
Table 1: There are no distances between 16m and 55m, which is a rather large "hole" and should be addressed and explained.

p14 l245: You never mention how much the system costs, but only say it is cheaper. This however is relevant information especially for people who aim to re-build their system as working time has to be factored in as well.

Author Response

Respond for review:

Title: Affordable robotic mobile mapping system based on LiDAR with additional rotating planar reflector

Turnitin Result: 8%. It referred several times to "bip.put.poznan.pl". Please explain.

Respond: This work is an extension of Michal Pelka PhD Thesis ”Automation of the multi-sensor system calibration for mobile robotics applications”. We rephrased some critical sections, thus we hope the manuscript is sufficient in current revision.

Result (Rating out of 60 (60=best)): 41 (68%)

(1-10(best))
Novelty: 6
Writing&Language: 2
Scientific Rigor: 8
Introduction&Literature: 9
Method&Results: 8
Discussion&Conclusion: 8

In short: The authors present a self-build lidar system for SLAM.

Overall:
In total the authors show in depth knowledge of ongoing research and development and propose a valuable idea for a low cost lidar system for SLAM. I am missing a cost evaluation though which should be provided to emphasise how cheap the system really is. Furthermore, the language needs to be improved significantly to allow easy readability. Please ask a native speaker to edit your manuscript or use a professional English editing service.

Respond: language is improved.

Please use "lidar" not "LiDAR" or anything else as "lidar" is a word in its own right based on the Oxford English dictionary and Merriam Webster Dictionary. (compare: radar)

Respond: Done.

The order of the introduction is a bit confused. You should state what current research is out in the area, explain relevant concepts that are very important for your paper or unknown to the target audience of the journal and then come to the problem that you want to solve and why it needs solving. You however have the aim somewhere in paragraph 2. Please rethink the order of paragraphs and also link them or easier transition for the reader between those.

Respond: We add some comments to address this issue.

Detail Comments:
page 5 line 108: You mention plane fitting without mentioning which algorithm you use. Please specify
Table 1: There are no distances between 16m and 55m, which is a rather large "hole" and should be addressed and explained.

Respond: Issue addressed.

p14 l245: You never mention how much the system costs, but only say it is cheaper. This however is relevant information especially for people who aim to re-build their system as working time has to be factored in as well.

Respond: Issue addressed.

Reviewer 4 Report

This paper proposed an Affordable robotic mobile mapping system based on LiDARwith additional rotating planar reflector compared to more expensive Velodyne VLP-LiDAR. Experience shows that the proposed solution can reach satisfactory accuracy and range. Here are some opinions. 

Advantages:

1. This paper has enough amount of work, including 3D geometrical calibration, mechanical design of mirror drive, an open-source project, which is good.

2. This paper is clear and easy to follow.

Cons:

1. Related work should add more latest papers and research in 2022 and 2023.

2. The idea of affordable LiDAR with rotating mirror or prism is not a new one, you should write your own contribution in introduction's end.

3. Figure 9 looks like point clouds with rotated reflector are lack shape and disordered, which is not a good performance to me, you should provide real ground-truth also. The caption location of Table 1 should be adjusted.

4. In evaluation or robotics match, are there some assessment indicators which to make people understand performance, like reference time, quality, etc.

Author Response

Respond for review:

This paper proposed an Affordable robotic mobile mapping system based on LiDARwith additional rotating planar reflector compared to more expensive Velodyne VLP-LiDAR. Experience shows that the proposed solution can reach satisfactory accuracy and range. Here are some opinions. 

Advantages:

1. This paper has enough amount of work, including 3D geometrical calibration, mechanical design of mirror drive, an open-source project, which is good.

2. This paper is clear and easy to follow.

Cons:

1. Related work should add more latest papers and research in 2022 and 2023.

Respond: was addressed this issue by expanding bibliography with relevant contributions in the field.

2. The idea of affordable LiDAR with rotating mirror or prism is not a new one, you should write your own contribution in introduction's end.

Respond: we addressed this comment in the end of introduction.

3. Figure 9 looks like point clouds with rotated reflector are lack shape and disordered, which is not a good performance to me, you should provide real ground-truth also. The caption location of Table 1 should be adjusted.

Respond: We clarified it and add new figure.

4. In evaluation or robotics match, are there some assessment indicators which to make people understand performance, like reference time, quality, etc.

Respond: we hope we clarified result by adding more comments.